# SAMF-YOLO: A self-supervised, high-precision approach for defect detection in complex industrial environments

**Jun Huang**[1,2]*, **Shamsul Arrieya Ariffin**[2,3], **Qiang Zhu**[1], **Wanting Xu**[2,4], **Qun Yang**[1]

**1** Faculty of Intelligent Manufacturing, Wuhu Institute of Technology, Anhui, China, **2** Faculty of Information Technology, City University Malaysia, Kuala Lumpur, Malaysia, **3** Faculty of Computing and Meta Technology, Sultan Idris Education University, Perak, Malaysia, **4** Faculty of Automotive and Aviation, Wuhu Institute of Technology, Anhui, China

* huangjun1207@whit.edu.cn

**Data availability statement:** All datasets and code files are available from the github database (https://github.com/Missing24ff/SAMF-YOLO/tree/master).

## Abstract

As object detection models grow in complexity, balancing computational efficiency and feature expressiveness becomes a critical challenge. To address this, we propose SAMF-YOLO, a novel model integrating three key components: SONet, BFAM, and FASFF-Head. The UniRepLKNet backbone, enhanced by the Star Operation, expands the feature space with high efficiency. FASFF-Head performs adaptive multi-scale feature fusion with minimal overhead, and the Bi-temporal Feature Aggregation Module (BFAM) strengthens the detection of small defects. Additionally, the Focaler-IoU loss improves bounding box regression for challenging object scales, and a self-supervised contrastive learning strategy enhances feature representation and model robustness without relying on labeled data. Experimental results demonstrate that SAMF-YOLO surpasses YOLOv11s with a 6.38% improvement in mAP@0.5 and a notable reduction in computational cost, confirming its superiority in accuracy, efficiency, and robustness. The code is released at https://github.com/Missing24ff/SAMF-YOLO.git.

## Introduction

Object detection has witnessed tremendous progress in recent years, largely propelled by deep learning techniques, especially convolutional neural networks (CNNs). Although models such as YOLO, SSD, and Faster R-CNN [1–3] have set benchmarks for performance and speed, they often face challenges in balancing computational efficiency with the capacity to learn rich, high-dimensional representations. Traditional convolutional backbones, despite their power, depend on complex hierarchical structures and explicit non-linear activation functions. This not only introduces substantial computational overhead but also requires extensive manual tuning for optimal performance [4,5].

To overcome these limitations, we propose UniRepLKNet, a novel backbone architecture integrated into the YOLOv11 framework. Leveraging the Star Operation, UniRepLKNet

**Funding:** This work is supported to JUN HUANG in part by the Key Nature Science Research Projects of Anhui Province under Grant 2022AH052187; in part by the General Program for the Training of Outstanding Young Teachers of Anhui Province under Grant YQYB2024126; in part by the Wuhu Institute of Technology Natural Science Research Key Projects under Grant wzyzrzd202409; in part by the Wuhu Institute of Technology Science and Technology Team Project under Grant wzykytd202208, and wzykjtd202204, and wzykjtd202202; in part by the Department of Education Skill Master Studio of Anhui Province under Grant 2023jnds002.The funders had no role in study design, data collection and analysis, decision to publish, or preparation of the manuscript.

**Competing interests:** The authors have declared that no competing interests exist.

generates high-dimensional features within a low-dimensional computational space, significantly enhancing feature extraction capabilities without incurring additional computational costs. By stacking multiple Star Operation layers [6], the feature space grows exponentially, thereby increasing the network's representational capacity. This design enables UniRepLKNet to maintain high efficiency while enriching feature dimensionality, offering a compelling alternative to traditional convolution-heavy backbones [7].

In addition, we introduce the FASFF-Head, an adaptive multi-scale fusion mechanism that improves detection robustness across varying object sizes [8]. Complementing this is the Bi-temporal Feature Aggregation Module (BFAM), which aggregates low-level textures and multi-level semantic cues to enhance detection precision, particularly for small or complex defects [9]. To further improve bounding box regression, we present the Focaler-IoU loss, designed to mitigate sample imbalance by emphasizing hard examples during training [10].

Extensive experiments demonstrate that SAMF-YOLO, which integrates these innovations, achieves notable improvements over existing methods in both detection accuracy and computational efficiency, making it a powerful and practical solution for real-time object detection in industrial applications [11,12].

## Related work

Object detection has been a key area of research within computer vision, and numerous methods have emerged to balance speed, accuracy, and efficiency. Early object detection models, such as R-CNN, achieved remarkable success by using region proposals and CNNs[13]. However, these methods were computationally expensive and slow, which led to the development of more efficient models like Fast R-CNN [14] and Faster R-CNN [15]. These models introduced the concept of region proposal networks (RPNs), which significantly improved detection speed and accuracy.

The YOLO series revolutionized real-time object detection by framing the task as a regression problem, allowing predictions to be made in a single network pass. YOLOv3 [16] and YOLOv4 [17] continued to improve upon the framework, incorporating additional techniques like multi-scale predictions and improved anchor box generation. YOLOv5 [18] further optimized the architecture, improving inference speed and detection performance.

In recent years, lightweight architectures have gained attention due to the increasing demand for real-time inference on resource-constrained devices. Models like MobileNetv2 [19] and EfficientNet [20] introduced depthwise separable convolutions and compound scaling techniques to improve computational efficiency without sacrificing accuracy. MobileNeXt [21] and MobileNetv3 [22] further optimized these techniques for edge devices, achieving significant improvements in both accuracy and efficiency.

Feature fusion techniques have also been explored extensively, with methods like Feature Pyramid Networks (FPN) [23] and BiFPN [24] providing robust multi-scale feature fusion. These networks combine features from different levels of the network to enhance object detection, especially for small or distant objects. The use of attention mechanisms has become increasingly popular, with Squeeze-and-Excitation Networks (SENet) [25] and Non-local Networks [26] improving feature selection by emphasizing important regions of the input data.

Moreover, the introduction of advanced loss functions like IoU-based losses [27] and Focal Loss [28] has significantly improved the performance of object detection networks, particularly for tasks involving small objects or imbalanced datasets. Despite these advancements, existing models still struggle with achieving both high computational efficiency and feature expressiveness. Our proposed UniRepLknet, combined with the FASFF-Head, BFAM, and

Focaler-IoU loss, offers a novel solution by focusing on efficiently learning implicit high-dimensional features and enhancing feature fusion while reducing computational complexity [29,30].

## Materials and methods

To address the limitations of existing models in the detection of complex industrial part defects, this study proposes an enhanced detection framework based on YOLOv11, named SAMF-YOLO. As shown in Fig 1, the model achieves significant performance improvements through three core innovative modules: First, a novel UniRepLknet backbone network is designed, which generates high-dimensional feature representations in low-dimensional computational space using the Star Operation. This effectively compensates for the feature loss caused by traditional pooling operations. The structure adopts a four-stage hierarchical architecture, with a dynamic feature dimension adjustment through a channel expansion factor, enabling fine-grained multi-scale feature extraction while maintaining computational efficiency. Second, to address the challenges of defect detection in complex scenarios, an effective module for filtering conflicting information and enhancing scale invariance (FASFF-HEAD) is proposed. This module can be trained via backpropagation and has minimal computational overhead, effectively solving the feature loss issue caused by cross-scale interactions. Lastly, to address the insufficiency of traditional initial convolutional feature capture, a Bi-temporal Feature Aggregation Module (BFAM) is introduced to replace the standard C3K2 blocks in the backbone network. BFAM gradually merges low-level texture information with multi-scale features, allowing the network to capture subtle changes and broader spatial relationships. After channel recalibration, these features are fused, improving the quality of the initial features while maintaining computational efficiency. Experimental results demonstrate that the Focaler-IoU loss function significantly enhances the model's detection performance for complex-shaped targets, improving mAP@0.5 by 6.38% compared to the baseline model. The technical highlights of each innovative module are as Fig 1 follows:

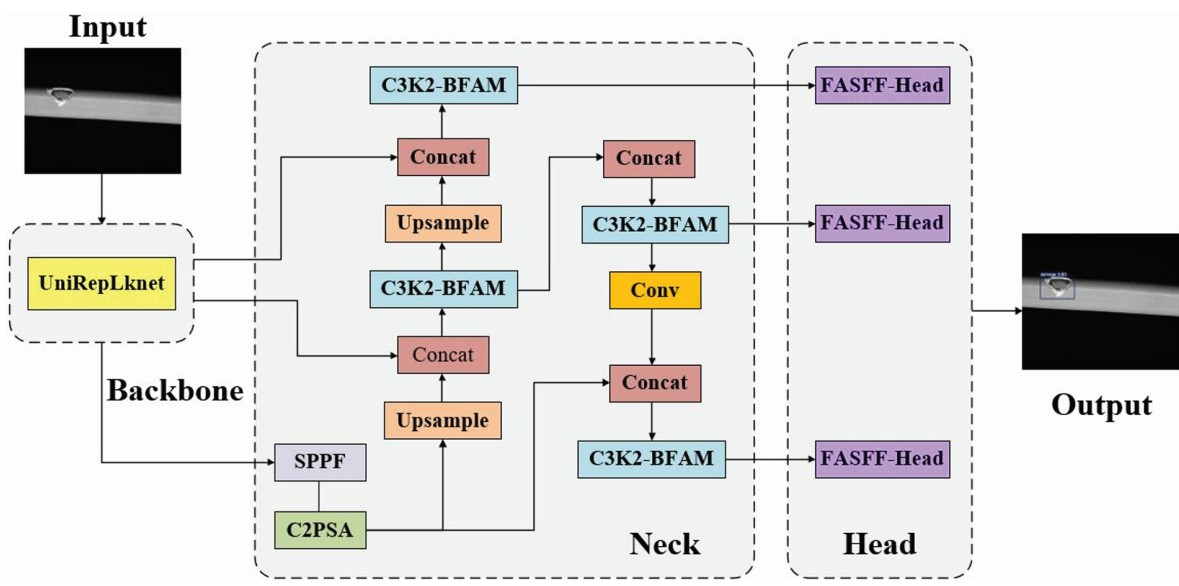

**Fig 1. Structure of the SAMF-YOLO model.**

## UniRepLknet

With the increasing complexity of object detection models, balancing computational efficiency and expressive power remains a major challenge. Traditional convolutional backbones rely on explicit non-linear activation functions and complex hierarchical structures to enhance feature extraction. However, these approaches introduce significant computational overhead and require meticulous manual optimization. To address this issue, we propose UniRepLknet, an innovative backbone that integrates the Star Operation into the YOLOv11 framework, offering a novel perspective on implicit high-dimensional representation learning.

In a single layer of a neural network, the star operation is typically expressed as:

$$\left(W_1^T X + B_1\right) * \left(W_2^T X + B_2\right) \tag{1}$$

which represents the fusion of two linearly transformed features through element-wise multiplication. To simplify the notation, we combine the weight matrices and bias terms into a single entity, defined as:

$$w_1, w_2, x \in \mathbb{R}^{(d+1)\times 1} \tag{2}$$

Where $d$ represents the number of input channels. This formulation can be naturally extended to accommodate multiple output channels, as follows:

$$W_1, W_2 \in \mathbb{R}^{(d+1)\times(d'+1)} \tag{3}$$

Thus, the star operation can be rewritten as:

$$
\begin{aligned}
w_1^{\mathrm{T}} x \star w_2^{\mathrm{T}} x &= \left(\sum_{i=1}^{d+1} w_1^i x^i\right) * \left(\sum_{j=1}^{d+1} w_2^j x^j\right) \\
&= \sum_{i=1}^{d+1}\sum_{j=1}^{d+1} w_1^i w_2^j x^i x^j \\
&= \underbrace{\alpha_{(1,1)} x^1 x^1 + \cdots + \alpha_{(4,5)} x^4 x^5 + \cdots + \alpha_{(d+1,d+1)} x^{d+1} x^{d+1}}_{(2+d)(1+d)/2 \text{ items}}
\end{aligned}
\tag{4}
$$

Here, we use $i,j$ to index the channels and define $\alpha$ as the coefficient for each term:

$$
\alpha_{(i,j)} = \begin{cases} w_1^i w_2^j & \text{if } i = j, \\ w_1^i w_2^j + w_1^j w_2^i & \text{if } i \neq j. \end{cases}
\tag{5}
$$

Except for $\alpha_{(d+1,\cdot)} x^{d+1} x$, each term maintains a nonlinear relationship with $x$, indicating that they correspond to independent, implicit high-dimensional feature dimensions. Consequently, while computations are performed within a $d$-dimensional space using the computationally efficient star operation, the resulting feature representation resides in an implicit feature space of dimension $\frac{(d+2)(d+1)}{2} \approx \left(\frac{\sqrt{d}}{2}\right)^2$.

Given that $d>2$, this characteristic effectively expands the feature dimensions without introducing additional computational overhead within a single layer. By stacking multiple layers of star operations, the implicit dimensions can grow exponentially in a recursive

manner, approaching infinity. Assuming the initial network layer has a width of $d$, applying a single star operation yields the following expression:

$$\sum_{i=1}^{d+1}\sum_{j=1}^{d+1} w_1^i w_2^j x^i x^j \tag{6}$$

This operation results in an implicit feature space of dimension $\mathbb{R}^{\left(\frac{d}{\sqrt{2}}\right)^2}$. Let $M_l$ denote the output of the $l$-th star operation, then we obtain:

$$\begin{cases} M_1 = \displaystyle\sum_{i=1}^{d+1}\sum_{j=1}^{d+1} w_{(1,1)}^i w_{(1,2)}^j x^i x^j, & M_1 \in \mathbb{R}^{\left(\frac{d}{\sqrt{2}}\right)^2} \\ M_l = \left(W_{l,1}^{\mathrm{T}} O_{l-1}\right) * \left(W_{l,2}^{\mathrm{T}} O_{l-1}\right), & M_l \in \mathbb{R}^{\left(\frac{d}{\sqrt{2}}\right)^{2^l}}, \quad l \ge 2 \end{cases} \tag{7}$$

After $l$ layers of computation, the feature space expands to a dimension of $\mathbb{R}^{\left(\frac{d}{\sqrt{2}}\right)^{2^l}}$. This means that by stacking only a few layers of star operations, the implicit feature dimension can be exponentially expanded, significantly enhancing the representation capacity.

To transform UniRepLknet into a self-supervised learning model, we adopt a contrastive learning approach aimed at learning effective feature representations through unsupervised tasks. Specifically, during training, positive and negative sample pairs are generated through data augmentation techniques, including random cropping (crop ratio between 0.6 and 1.0 of the original image), horizontal flipping (with a probability of 0.5), random rotation (within the range of −15° to +15°), color jittering (brightness, contrast, saturation, and hue adjusted with a strength factor of 0.4), grayscale transformation (with a probability of 0.2), Gaussian blur (applied with a kernel size of 3 and sigma in the range [0.1, 2.0]), and random erasing (with an area ratio ranging from 2% to 10% of the image), as depicted in Fig 2. Positive sample pairs are formed by applying two distinct augmentations to the same original image, thereby producing two different but semantically similar views. Negative samples, conversely, are derived from different images, ensuring that the learned representation distinguishes clearly between varied object instances.

Assuming the input data consists of augmented pairs of samples $(x_i, x_j)$, their respective high-dimensional feature vectors $(z_i, z_j)$ are extracted via the UniRepLknet backbone utilizing the Star Operation. To measure similarity between the feature vectors, we employ the cosine similarity metric as defined:

$$\mathrm{sim}(\mathbf{z}_i, \mathbf{z}_j) = \frac{\mathbf{z}_i^{\mathrm{T}} \mathbf{z}_j}{|\mathbf{z}_i||\mathbf{z}_j|} \tag{8}$$

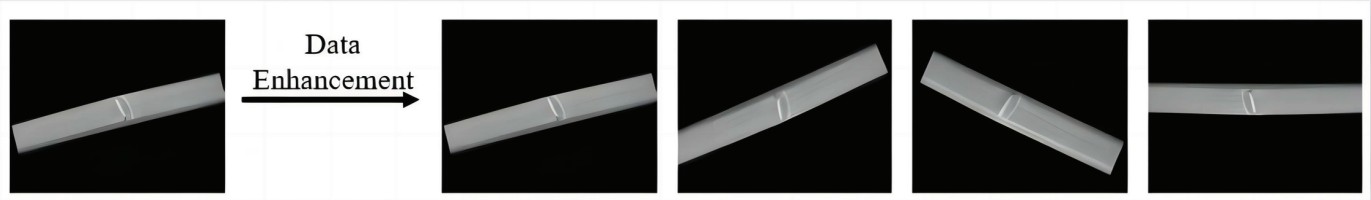

**Fig 2. Data augmentation.**

For the optimization objective, we employ the widely-adopted InfoNCE loss as our contrastive loss function. This loss function aims to maximize similarity for positive pairs while minimizing similarity for negative pairs:

$$\mathcal{L}\text{contrastive} = -\log \frac{\exp\big(\text{sim}(z_i, z_j)/\tau\big)}{\sum k = 1^N \exp\big(\text{sim}(z_i, z_k)/\tau\big)} \tag{9}$$

Here, $\tau$ represents the temperature parameter, which scales the distribution of similarity scores and is empirically set to 0.07 in our experiments. During training, we adopt the SGD optimizer with an initial learning rate of 0.01, momentum of 0.937, and weight decay of $5 \times 10^{-4}$, accompanied by a cosine annealing schedule to gradually adjust the learning rate over epochs.

Combining the Star Operation in UniRepLknet with contrastive learning, the final self-supervised loss function integrates the contrastive loss and an L2 regularization term to prevent feature space collapse and enhance the model's generalization capability:

$$\mathcal{L}\text{self} - \text{supervised} = \mathcal{L}\text{contrastive} + \lambda \mathcal{L}_{\text{regularization}} \tag{10}$$

where $\lambda$ is empirically set to $1 \times 10^{-4}$ to balance the impact of the regularization term.

Given the unique advantage of UniRepLknet–which performs computations in a low-dimensional space while generating high-dimensional features–we have identified its practical value in efficient network architectures. Therefore, we integrate UniRepLknet as a proof-of-concept model into the YOLOv11 backbone, where it demonstrates outstanding performance, highlighting the effectiveness of the Star Operation, as Fig 3 shows.

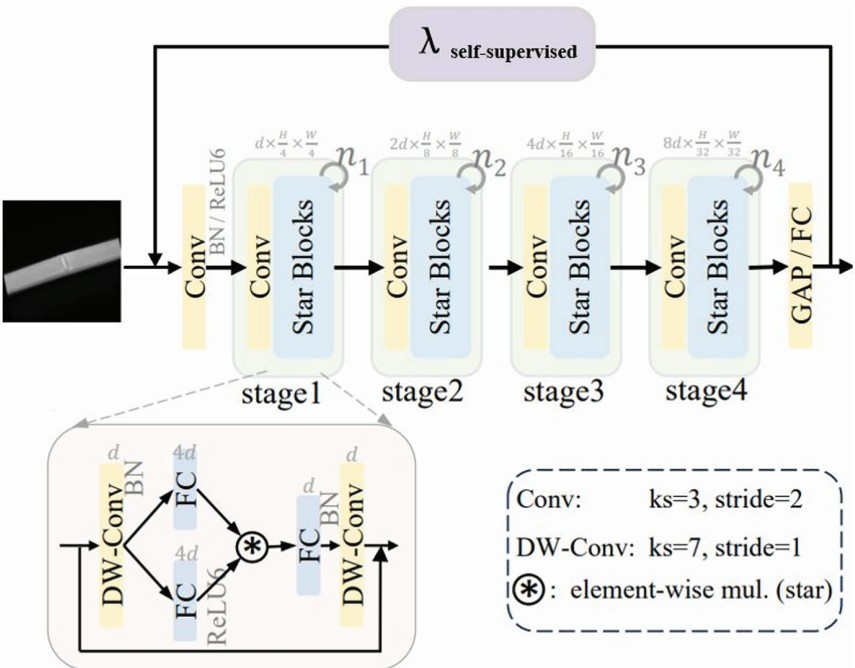

**Fig 3. UniRepLknet architecture overview.**

UniRepLknet adopts a four-stage hierarchical architecture, where downsampling is performed through convolutional layers, and feature extraction is carried out using modified blocks. To enhance efficiency, Batch Normalization (BatchNorm) replaces Layer Normalization, positioned after depthwise convolution, enabling fusion during inference. Inspired by MobileNeXt, we introduce an additional depthwise convolution at the end of each module. The channel expansion factor is consistently set to 4, while the network width doubles at each stage. The architecture follows the design principles of MobileNetv2, replacing the GELU activation function in the demonstration blocks with ReLU6, further improving computational efficiency. The overall UniRepLknet framework is illustrated in the accompanying diagram. By simply adjusting the number of blocks and input embedding channels, we construct different scales of UniRepLknet, optimizing the YOLOv11 backbone to achieve a balance between efficiency and performance.

## FASFF-Head

Unlike traditional feature fusion methods, we introduce FASFF-HEAD, which integrates Four-Head Adaptive Spatial Feature Fusion (FASFF) directly into the YOLOv11 detection head. This approach enables more efficient multi-scale feature fusion while addressing feature loss caused by cross-scale interactions. The key innovation of FASFF-HEAD lies in its adaptive spatial feature fusion mechanism, which effectively filters conflicting information and enhances scale invariance. The FASFF method is model-agnostic, trainable via backpropagation, and introduces minimal computational overhead, making it a practical enhancement for existing object detection frameworks, as Fig 4 shows. The Key Innovations of FASFF-HEAD:

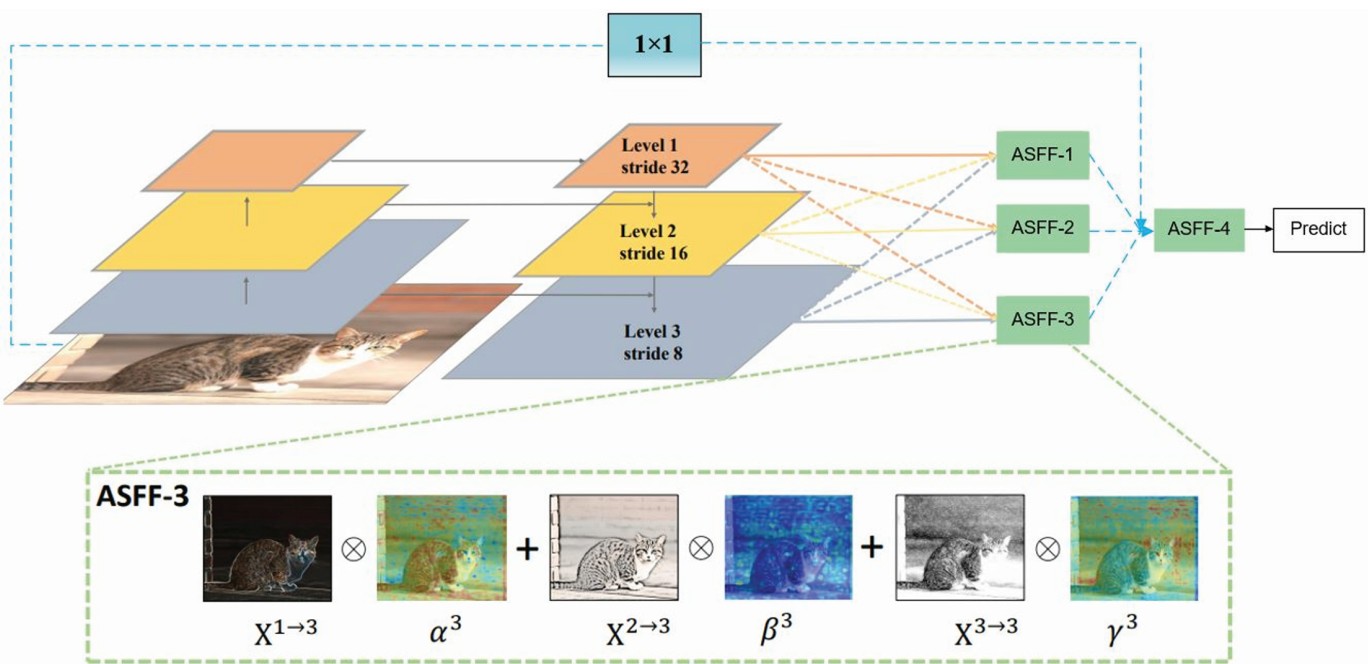

**Fig 4. Illustration of the adaptively spatial feature fusion mechanism.**

1. **Adaptive Spatial Feature Fusion:** We propose a novel pyramid feature fusion strategy that spatially filters conflicting information and suppresses inconsistencies between multi-scale features.
2. **Enhanced Scale Invariance:** The FASFF strategy significantly improves feature scale invariance, leading to more accurate object detection.
3. **Low Inference Overhead:** While improving detection performance, FASFF introduces negligible additional computational cost during inference.

These innovations make FASFF-HEAD a significant advancement in the field of single-shot object detection, particularly in improving the ability to handle objects of varying scales. Figure A illustrates the working principle of the Adaptive Spatial Feature Fusion (FASFF) mechanism, specifically designed for single-shot object detection.

In this structure, features at different levels (represented by layers of different colors) first undergo downsampling or upsampling according to their respective strides to ensure uniform spatial dimensions across all feature maps. Level 1, Level 2, Level 3, and Level 4 correspond to different levels in the feature pyramid, each with distinct spatial resolutions. ASFF-1, ASFF-2, ASFF-3, and ASFF-4 represent different levels where the FASFF mechanism is applied to achieve feature fusion.

In the zoomed-in section of FASFF-4, we observe that feature maps from other levels $(x_1 \rightarrow 1, x_2 \rightarrow 2, x_3 \rightarrow 3)$ are resized to match the spatial dimensions of the fourth level $(x_4 \rightarrow 4)$. These resized features are then adaptively weighted using learned importance maps before being fused into a final refined feature representation $(\hat{y}^4)$, which is used for prediction. Let $x_{ij}^{(n \rightarrow l)}$ denote the feature vector at the position $(i,j)$ on the feature maps resized from level $n$ to level $l$. We propose to fuse the features at the corresponding level $l$ as follows:

$$\mathbf{y}_{ij}^l = \alpha_{ij}^l \cdot \mathbf{x}_{ij}^{1 \rightarrow l} + \beta_{ij}^l \cdot \mathbf{x}_{ij}^{2 \rightarrow l} + \gamma_{ij}^l \cdot \mathbf{x}_{ij}^{3 \rightarrow l} + \delta_{ij}^l \cdot \mathbf{x}_{ij}^{4 \rightarrow l} \tag{11}$$

where $y_{ij}^l$ denotes the $(i,j)$-th vector of the output feature maps $y^l$ among channels. $\alpha_{ij}^l, \beta_{ij}^l, \gamma_{ij}^l$, and $\delta_{ij}^l$ refer to the spatial importance weights for the feature maps at three different levels to level $l$, which are adaptively learned by the network. Note that $\alpha_{ij}^l, \beta_{ij}^l, \gamma_{ij}^l$, and $\delta_{ij}^l$ can be simple scalar variables, shared across all the channels. Inspired by [31], we enforce the constraint $\alpha_{ij}^l + \beta_{ij}^l + \gamma_{ij}^l + \delta_{ij}^l = 1$ and $\alpha_{ij}^l, \beta_{ij}^l, \gamma_{ij}^l, \delta_{ij}^l \in [0, 1]$, and define:

$$\alpha_{ij}^l = \frac{e^{\lambda_{\alpha_{ij}}^l}}{e^{\lambda_{\alpha_{ij}}^l} + e^{\lambda_{\beta_{ij}}^l} + e^{\lambda_{\gamma_{ij}}^l} + e^{\lambda_{\delta_{ij}}^l}} \tag{12}$$

Here, $\alpha_{ij}^l, \beta_{ij}^l, \gamma_{ij}^l$, and $\delta_{ij}^l$ are defined by using the softmax function with $\lambda_{\alpha_{ij}}^l, \lambda_{\beta_{ij}}^l, \lambda_{\delta_{ij}}^l$, and $\lambda_{\gamma_{ij}}^l$ as control parameters, respectively. We use $1 \times 1$ convolution layers to compute the weight scalar maps $\lambda_\alpha^l, \lambda_\beta^l, \lambda_\gamma^l$, and $\lambda_\delta^l$ from $x^{(1 \rightarrow l)}, x^{(2 \rightarrow l)}, x^{(3 \rightarrow l)}$, and $x^{(4 \rightarrow l)}$, respectively. These weight maps can thus be learned through standard back-propagation.

With this method, the features at all the levels are adaptively aggregated at each scale. The outputs $\{y^1, y^2, y^3, y^4\}$ are used for object detection following the same pipeline of YOLOv11. By incorporating four-head feature fusion into the detection pipeline, FASFF-HEAD achieves weighted fusion across multiple feature levels. Compared to traditional three-head fusion methods, this approach further enhances detection accuracy, especially in complex environments and for objects of varying scales.

## C3K2-BFAM

In the original YOLOv11 framework, the C3K2 layer primarily focuses on capturing mid-level features but lacks a comprehensive integration of low-level details and multilevel information. While the C3K2 layer captures important features, it struggles to effectively combine fine-grained details (such as texture and color) with higher-level context, which is crucial for detecting small, intricate industrial defects.

To address this limitation, we introduce the Bi-temporal Feature Aggregation Module (BFAM) within the C3K2 layer, as shown in the Fig 5. BFAM progressively merges low-level texture information with multilevel features, allowing the network to capture both subtle changes and broader spatial relationships. This fusion improves the model's ability to detect small, complex defects, which is essential for industrial part defect detection.

To effectively extract fine-grained features while preserving spatial relationships, we apply channel splicing to the input bitemporal features $(F_i, f_i)$. Four parallel dilated convolutions $(3 \times 3)$ with varying dilation rates $(1, 2, 3, 4)$ and group size $c$ (representing the number of channels) are employed. This strategy enables the model to capture change regions at different scales through diverse receptive fields, all while maintaining the spatial coherence of the features via grouped convolutions. The outputs from these convolutions are then concatenated along the channel dimension, followed by a $1 \times 1$ convolution block to reduce the channel dimensions. Finally, feature refinement is achieved using the SimAM attention mechanism [32]. The corresponding equations are as follows:

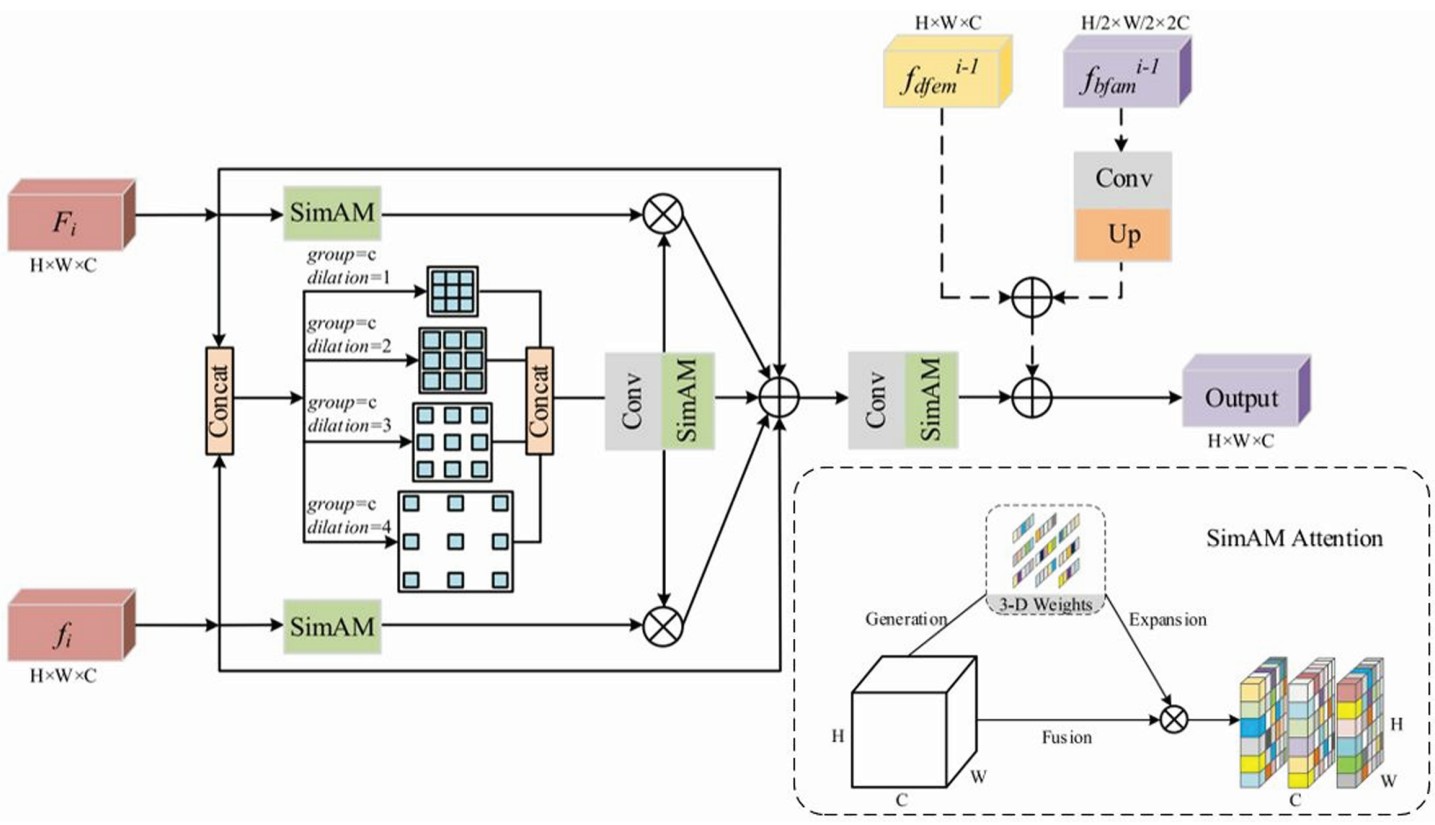

**Fig 5. Details of the BFAM.**

$$f_{3\times3}^{d,g} = \text{Conv}_{3\times3}^{d,g}\left(\text{Concat}(F_i, f_i)\right) d \in \{1,2,3,4\}, g = c \tag{13}$$

$$f_{cat} = \text{Conv}_{1\times1}\left(\text{Concat}\left(f_{3\times3}^{1,c}, f_{3\times3}^{2,c}, f_{3\times3}^{3,c}, f_{3\times3}^{4,c}\right)\right) \tag{14}$$

$$\hat{f}_{\text{cat}} = \text{SimAM}(f_{\text{cat}}) \tag{15}$$

Let $d$ represent the dilation rate, $g$ the group size, and *Concat* the channel concatenation operation. Considering the shared characteristics within the bitemporal features, this approach enhances the precision of low-level texture information. We compute the importance of pixel-level features from different temporal inputs using the SimAM attention mechanism. Subsequently, the extracted common feature $\hat{f}_{\text{cat}}$ is multiplied by the respective temporal features $(F_i, f_i)$ to measure their similarity. The equations are as follows:

$$\hat{F}_i = \text{SimAM}(F_i) \times \hat{f}_{\text{cat}} \tag{16}$$

$$\hat{f}_i = \text{SimAM}(f_i) \times \hat{f}_{\text{cat}} \tag{17}$$

Next, we combine the low-level detail features $(\hat{F}_i, \hat{f}_i, \hat{f}_{\text{cat}})$ with the advanced change feature $f_{(i-1)}^{\text{dfem}}$. The final aggregated feature is generated using the SimAM attention mechanism. As the CNN model deepens, low-level information gradually diminishes. To retain sufficient texture details in the deeper layers, we introduce residual connections. The equations are as follows:

$$f_{\textbf{bfam}}^i = \begin{cases} \text{SimAM}\left(\text{Conv}_{3\times3}(\hat{F}_i + \hat{f}_i + \hat{f}_{\textbf{cat}})\right) & \text{if } i = 1, \\ \text{SimAM}\left(\text{Conv}_{3\times3}(\hat{F}_i + \hat{f}_i + \hat{f}_{\textbf{cat}} + f_{\textbf{dfem}}^{i-1})\right) & \text{if } i = 2,3,4, \\ +\text{Up}\left(\text{Conv}_{1\times1}(f_{\textbf{hfam}}^{i-1})\right) \end{cases} \tag{18}$$

We incorporated multilevel features through the use of a BFAM to improve the spatial texture information in the change region by integrating various receptive fields. Additionally, these features were fused with high-level semantic information to provide a more precise and holistic understanding of the change.

## Focaler-IoU

In various object detection tasks, the problem of imbalanced samples is prevalent. Samples can be classified into hard samples and easy samples according to the difficulty of object detection. From the perspective of target scale analysis, general detection targets can be regarded as easy samples, while extremely small targets are considered hard samples due to the great difficulty in their precise positioning.

For detection tasks dominated by easy samples, paying attention to easy samples during the bounding box regression process can contribute to the improvement of detection performance. Conversely, for detection tasks with a high proportion of hard samples, it is necessary to focus on the bounding box regression of hard samples.

To focus on different regression samples for various detection tasks, we reconstruct the Intersection over Union loss (IoU loss) using the linear interval mapping method [33]. This is conducive to enhancing the boundary regression effect. The formula is as follows:

$$IoU^{focaler} = \begin{cases} 0, & \text{if } IoU < d \\ \frac{IoU - d}{u - d}, & \text{if } d \ll IoU \ll u \\ 1, & \text{if } IoU > u \end{cases} \tag{19}$$

Where $IoU^{focaler}$ is the reconstructed Focaler-IoU, IoU is the original IoU value, and $[d, u] \in [0, 1]$. By adjusting the values of $d$ and $u$, we can make $IoU^{focaler}$ focus on different regression samples. Its loss is defined below:

$$L_{Focaler-IoU} = 1 - IoU^{focaler} \tag{20}$$

For general object detection tasks, placing appropriate emphasis on easy samples proves beneficial in boosting overall detection performance; conversely, in scenarios where extremely small targets constitute a significant proportion, directing attention to hard samples markedly enhances the precision of boundary localization. Subsequent experiments and analyses further substantiate that Focaler-IoU not only preserves detection accuracy but also effectively mitigates sample imbalance in bounding box regression, thereby paving the way for adaptive regression strategies tailored to varying target scales and detection contexts.

## Experimental results and analyses experimental environments and dataset

As shown in Table 1, the deep learning environment and framework used in the experiments have been consistently applied to all tested networks with identical configurations. Detailed hyper-parameters listed in Table 2 were uniformly used across all models evaluated, ensuring fairness and consistency throughout the experimental comparisons.

To comprehensively evaluate the robustness and generalization capability of the SAMF-YOLO model, two distinct datasets were employed. The first dataset, self-collected, consists of solar heating tape defect images, comprising 907 images for training and 227 images for validation. This dataset covers diverse defect types, including wrinkles, dents, scratches, punctures, damage, and dirt, providing varied scenarios to rigorously test the model's ability to generalize across different conditions and anomalies.Additionally, the publicly available NEU-DET [34] dataset was utilized to further validate the effectiveness and performance of our proposed approach. NEU-DET focuses specifically on steel surface defects, containing 1,800 grayscale images classified into six typical defect categories: crazing, patches, inclusion, pitted surface, rolled-in scale, and scratches. The incorporation of this standard dataset further demonstrates the model's adaptability and effectiveness in broader industrial defect detection tasks.

**Table 1. Experimental environment configuration.**

| Items | Description |
| --- | --- |
| Operating system | Windows |
| CPU | AMD 7800X |
| GPU | NVIDIA 4080 Super |
| GPU acceleration libraries | CUDA 12.0 |
| Graphics memory | 16GB |
| Python | 3.11 |
| Frameworks | Pytorch 2.2.2 |

**Table 2. Network hyperparameters.**

| Training parameters | Values |
|---|---|
| Image size | $640 \times 640$ |
| Epochs | 250 |
| Batch size | 64 |
| Learning rate | 0.01 |
| Optimizer | SGD |
| Momentum | 0.937 |
| Weight decay | 0.0005 |

To ensure efficient training and evaluation, the experiments were conducted on high-performance hardware, configured with an NVIDIA RTX 4080 Super GPU and CUDA 12.1 to enhance computational efficiency. The software environment is based on Python 3.11, with model training and evaluation carried out using the PyTorch deep learning framework. Additionally, to improve the consistency of training and evaluation, the datasets were split into 70% for training, 20% for validation, and 10% for testing. The dataset was meticulously annotated to capture complex scenarios such as occlusion and texture variations. All images were resized to 640×640 before being fed into the model to maintain consistency and reduce computational load.

With these two datasets and robust hardware support, the SAMF-YOLO model is capable of efficient defect detection in a wide range of real-world environments, ensuring its broad applicability and high efficiency.

## Evaluation metrics

This study evaluates object detection performance through Precision, Recall, mAP, GFLOPs, and model parameters. Classification outcomes are defined by True Positives (TP), False Positives (FP), False Negatives (FN), and True Negatives (TN), providing a comprehensive view of accuracy and efficiency. Precision reflects the ability to correctly identify positive samples. High precision means fewer false positives, while low precision indicates a higher rate of incorrect positive predictions. It is expressed as follows:

$$P = \frac{TP}{TP + FP} \tag{21}$$

Recall quantifies the proportion of actual positive samples correctly identified by the classifier. A high recall indicates that most positive instances are detected, while a low recall suggests many positive instances are missed. It is expressed as follows:

$$R = \frac{TP}{TP + FN} \tag{22}$$

Average Precision (AP) measures object detection performance by computing the area under the Precision-Recall curve across different thresholds. A higher AP value signifies better detection accuracy. It is calculated as:

$$AP = \int_0^1 P \mathrm{d}R \tag{23}$$

Mean Average Precision (mAP) averages the AP values across all classes in multi-class tasks. mAP@0.5, a common benchmark, is computed at an IoU threshold of 0.5 to assess detection performance. It is expressed as:

$$mAP50 = \frac{1}{N} \sum_{i=1}^{N} AP_i \tag{24}$$

Where $N$ represents the total number of classes, and $AP_i$ denotes the average precision for the $i$-th class.

To provide a more comprehensive evaluation, mAP@0.5:0.95 averages AP across multiple IoU thresholds ranging from 0.5 to 0.95 with a step size of 0.05. This metric reflects both localization accuracy and robustness. It is defined as:

$$mAP_{0.5:0.95} = \frac{1}{T} \sum_{t=1}^{T} \left( \frac{1}{N} \sum_{i=1}^{N} AP_i^{@t} \right) \tag{25}$$

Where $T$ is the number of IoU thresholds (typically 10), and $AP_i^{@t}$ denotes the average precision for the $i$-th class at the $t$-th IoU threshold.

GFLOPs (Giga Floating Point Operations) quantify the computational complexity of a deep learning model. Specifically, they measure the number of floating-point operations required by the model, expressed in billions. A higher GFLOPs value indicates a more complex model that demands greater computational resources, resulting in longer training and inference times. This metric is essential for evaluating the efficiency and scalability of a model, especially when deploying it on limited hardware.

The number of parameters refers to the learnable weights and biases within a model, which directly impacts its capacity for learning complex features. While increasing the number of parameters can enhance the model's ability to capture intricate patterns in the data, it also introduces higher resource requirements. More parameters lead to increased memory usage for storage, higher computational costs during training and inference, and potentially longer training times. Striking a balance between model complexity and resource efficiency is key for optimizing performance and ensuring practical deployment.

## Ablation experiments

To evaluate the performance gain obtained by enlarging the implicit feature space with the *Star Operation*, we varied only its feature-dimension expansion coefficient $\gamma$, keeping every other architectural component and training hyper-parameter unchanged. As reported in Table 3, setting $\gamma$ = 1× lifts mAP@0.5 from 67.32% to 69.10%–a 1.78 pp increase–at the cost of just +0.6 M trainable parameters. Raising the coefficient to $\gamma$ = 2× yields a better trade-off: mAP@0.5 reaches 70.24% (a 2.92 pp gain over the baseline) while the model remains compact at 2.6 M parameters. Further widening to $\gamma$ = 4× provides only an additional 0.56 pp but nearly doubles the parameter budget, indicating sharply diminishing returns. Consequently, $\gamma$ = 2× is adopted as the default configuration in all subsequent experiments.

To evaluate the effectiveness of each proposed module in SAMF-YOLO, we conducted a comprehensive ablation study as shown in Table 4, focusing on the contributions of the Stars Operation Net (So-Net), Bi-temporal Feature Aggregation Module (BFAM), Adaptive Spatial Feature Fusion Head (FASFF), and the Focaler-IoU loss function.Using So-Net as the backbone alone yields a strong baseline, reaching 70.24% mAP@0.5 and 34.3% mAP@0.5:0.95,

**Table 3. The Star Operation feature–dimension expansion coefficient $\gamma$.**

| Expansion coeff. $\gamma$ | Precision (%) | mAP@0.5 (%) | mAP@0.5:0.95 (%) | Params |
|---|---|---|---|---|
| None (baseline) | 72.3 | 67.3 | 33.7 | 1.3 M |
| 1× | 72.3 | 69.1 | 34.1 | 1.9 M |
| 2× | 72.4 | 70.2 | 34.3 | 2.6 M |
| 4× | 73.1 | 70.8 | 34.6 | 4.7 M |

**Table 4. Model performance comparison.**

| So-Net | BFAM | FASFF | Focaler-IoU | Precision % | mAP@0.5 % | mAP@0.5:0.95 % | FPS |
|---|---|---|---|---|---|---|---|
| ✓ | × | × | × | 72.4 | 70.24 | 34.3 | 81.2 |
| ✓ | ✓ | × | × | 72.9 | 72.4 | 35.6 | 79.4 |
| ✓ | ✓ | ✓ | × | 73.2 | 75.1 | 36.8 | 78.6 |
| ✓ | ✓ | ✓ | ✓ | 76.5 | 75.7 | 37.4 | 77.9 |

with a high inference speed of 81.2 FPS. This demonstrates its ability to extract rich multi-scale features while maintaining efficiency.Introducing the BFAM module, which captures complementary temporal-spatial patterns, leads to notable improvements: Precision increases to 72.9%, mAP@0.5 rises to 72.4%, and mAP@0.5:0.95 improves to 35.6%, with only a slight FPS drop to 79.4.Adding the FASFF module to the detection head further enhances the results, especially for small-object detection. The model achieves 75.1% mAP@0.5 and 36.8% mAP@0.5:0.95, with only a minor speed trade-off (78.6 FPS), demonstrating FASFF's strength in adaptive spatial feature fusion.Finally, replacing the standard CIoU loss with Focaler-IoU yields the best overall performance: Precision reaches 76.5%, mAP@0.5 peaks at 75.7%, and mAP@0.5:0.95 rises to 37.4%, while maintaining high inference speed at 77.9 FPS.Overall, each component of SAMF-YOLO contributes significantly to performance, validating the design's effectiveness in building an accurate and efficient industrial defect detection framework.

The results in Table 5 demonstrate that our proposed Focaler-IoU loss function consistently outperforms existing IoU-based variants across all key evaluation metrics. It achieves the highest mAP@0.5 of 75.7% and the best mAP@0.5:0.95 of 37.4%, indicating superior localization performance across a broad IoU threshold range. In addition, it delivers the highest precision (76.5%) and recall (74.8%), showcasing a balanced improvement in both confidence and completeness of detections.Compared to traditional losses such as IoU (72.0%, 34.3%), GIoU (73.2%, 34.9%), and CIoU (73.9%, 35.8%), Focaler-IoU provides a clear edge,

**Table 5. Performances of different loss functions.**

| Loss function | Precision (%) | Recall (%) | mAP@0.5 (%) | mAP@0.5:0.95 (%) |
|---|---|---|---|---|
| IoU | 70.6 | 71.3 | 72.0 | 34.3 |
| GIoU | 71.8 | 72.4 | 73.2 | 34.9 |
| DIoU | 71.1 | 72.0 | 73.0 | 34.6 |
| EIoU | 71.4 | 72.2 | 73.5 | 35.2 |
| SIoU | 72.2 | 73.3 | 74.1 | 36.1 |
| CIoU | 71.9 | 72.7 | 73.9 | 35.8 |
| WIoU | 71.0 | 72.2 | 73.0 | 35.4 |
| Ours | 76.5 | 74.8 | 75.7 | 37.4 |

particularly in challenging scenarios where precise bounding box regression is critical,as Fig 6 shows.

This performance gain is particularly valuable in industrial defect detection tasks, where small and irregularly shaped objects require more refined localization. By emphasizing informative samples and adapting better to varying object geometries, Focaler-IoU enhances both detection robustness and reliability, making it a key contributor to the overall effectiveness of the SAMF-YOLO framework.

## Comparative experiment

In this study, we conducted a comprehensive evaluation of the proposed SAMF-YOLO algorithm against mainstream object detection models—including YOLOv5s [35], YOLOv6n [36], YOLOv7-tiny [37], YOLOv8n, RT-DETR-R18 [38], YOLOv9s [39], YOLOv10s [40], and YOLOv11s.

As shown in Table 6, on the solar heating tape surface defect dataset, SAMF-YOLO achieves the best trade-off between detection accuracy and computational efficiency. Compared with YOLOv5s, our model improves Precision from 60.42% to 76.5% and mAP@0.5 from 63.11% to 75.7%, while reducing GFLOPs from 15.8 to 12.4, demonstrating the efficiency of the UniRepLKNet backbone.Against YOLOv6n, SAMF-YOLO increases mAP@0.5 by 14.56% with only a marginal increase in computation (12.4 vs. 11.8 GFLOPs), validating

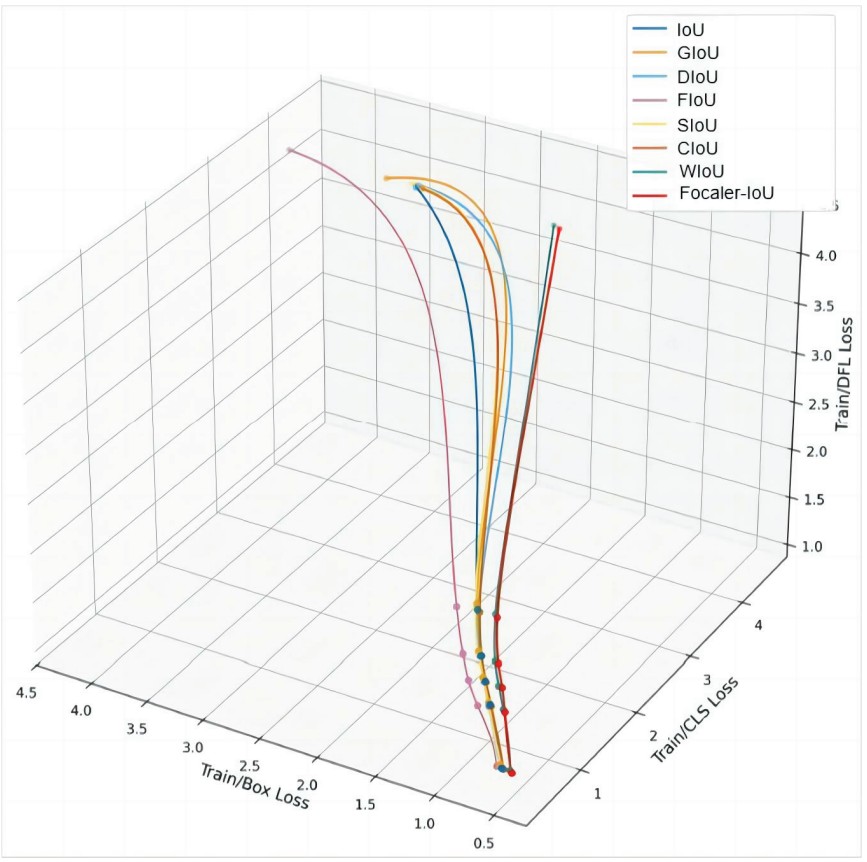

**Fig 6. Performances of different loss functions.**

**Table 6. Performance comparison on the solar heating tape surface defect dataset.**

| Model | Precision (%) | mAP@0.5 (%) | mAP@0.5:0.95 (%) | GFLOPs | FPS |
|---|---|---|---|---|---|
| YOLOv5s | 60.42 | 63.11 | 31.2 | 15.8 | 82.4 |
| YOLOv6n | 58.77 | 61.14 | 30.5 | 11.8 | 88.3 |
| YOLOv7-tiny | 65.43 | 63.49 | 31.9 | 13.0 | 84.6 |
| YOLOv8n | 61.32 | 64.32 | 32.5 | 8.1 | 91.7 |
| RT-DETR-R18 | 61.89 | 65.28 | 33.4 | 58.6 | 46.2 |
| YOLOv9s | 68.21 | 67.14 | 34.2 | 26.7 | 58.9 |
| YOLOv10s | 71.43 | 66.93 | 32.4 | 24.4 | 61.3 |
| YOLOv11s | 72.29 | 67.32 | 33.7 | 21.3 | 64.7 |
| Ours | 76.5 | 75.7 | 37.4 | 12.4 | 77.9 |

the lightweight design of the C3K2-BFAM module. Compared with YOLOv7-tiny, our model surpasses it by 12.21% in mAP@0.5 and reduces computational cost by 0.6 GFLOPs, confirming the advantages of Focaler-IoU and the overall efficient architecture.When compared to YOLOv8n, SAMF-YOLO achieves a 15.18% higher Precision and 11.38% higher mAP@0.5, while using only 12.4 GFLOPs versus 8.1, highlighting the robustness of the FASFF-Head in handling complex spatial features. Furthermore, compared to RT-DETR-R18, SAMF-YOLO improves mAP@0.5 by 10.42% and reduces GFLOPs by nearly 79%, making it substantially more suitable for real-time applications.Even against the latest models such as YOLOv9s, YOLOv10s, and YOLOv11s, SAMF-YOLO maintains superiority in both accuracy and efficiency—offering higher mAP@0.5 while reducing GFLOPs by 53.6%, 49.2%, and 41.8%, respectively. Notably, it sustains a real-time inference speed of 77.9 FPS, striking an ideal balance between performance and speed.

On the NEU-DET dataset (Table 7), SAMF-YOLO again demonstrates excellent generalization ability, achieving the highest scores across all key metrics: Precision of 79.46%, mAP@0.5 of 76.32%, and mAP@0.5:0.95 of 39.17%. Despite its lightweight architecture (12.4 GFLOPs), the model runs at a high-speed 85.7 FPS, confirming its robustness and versatility for diverse industrial defect detection tasks.

## Detection results and analysis

As shown in the Fig 7, the object detection results of different models for images of solar heating tapes with various defects are presented. The first column shows the original images, the second column presents the detection results of the baseline model YOLOv11s, and the third column shows the results of the improved SAMF-YOLO model. The first row displays

**Table 7. Performance comparison on the NEU-DET dataset.**

| Model | Precision (%) | mAP@0.5 (%) | mAP@0.5:0.95 (%) | GFLOPs | FPS |
|---|---|---|---|---|---|
| YOLOv5s | 72.14 | 70.87 | 35.02 | 15.8 | 94.5 |
| YOLOv6n | 70.36 | 69.45 | 33.94 | 11.8 | 99.6 |
| YOLOv7 | 74.28 | 71.92 | 37.20 | 13.0 | 96.8 |
| YOLOv8n | 71.45 | 70.38 | 35.71 | 8.1 | 105.3 |
| RT-DETR-R18 | 73.01 | 72.41 | 36.85 | 58.6 | 49.7 |
| YOLOv9s | 76.34 | 73.26 | 37.92 | 26.7 | 65.4 |
| YOLOv10s | 77.18 | 74.08 | 38.36 | 24.4 | 68.2 |
| YOLOv11s | 78.03 | 74.62 | 38.74 | 21.3 | 70.1 |
| Ours | 79.46 | 76.32 | 39.17 | 12.4 | 85.7 |

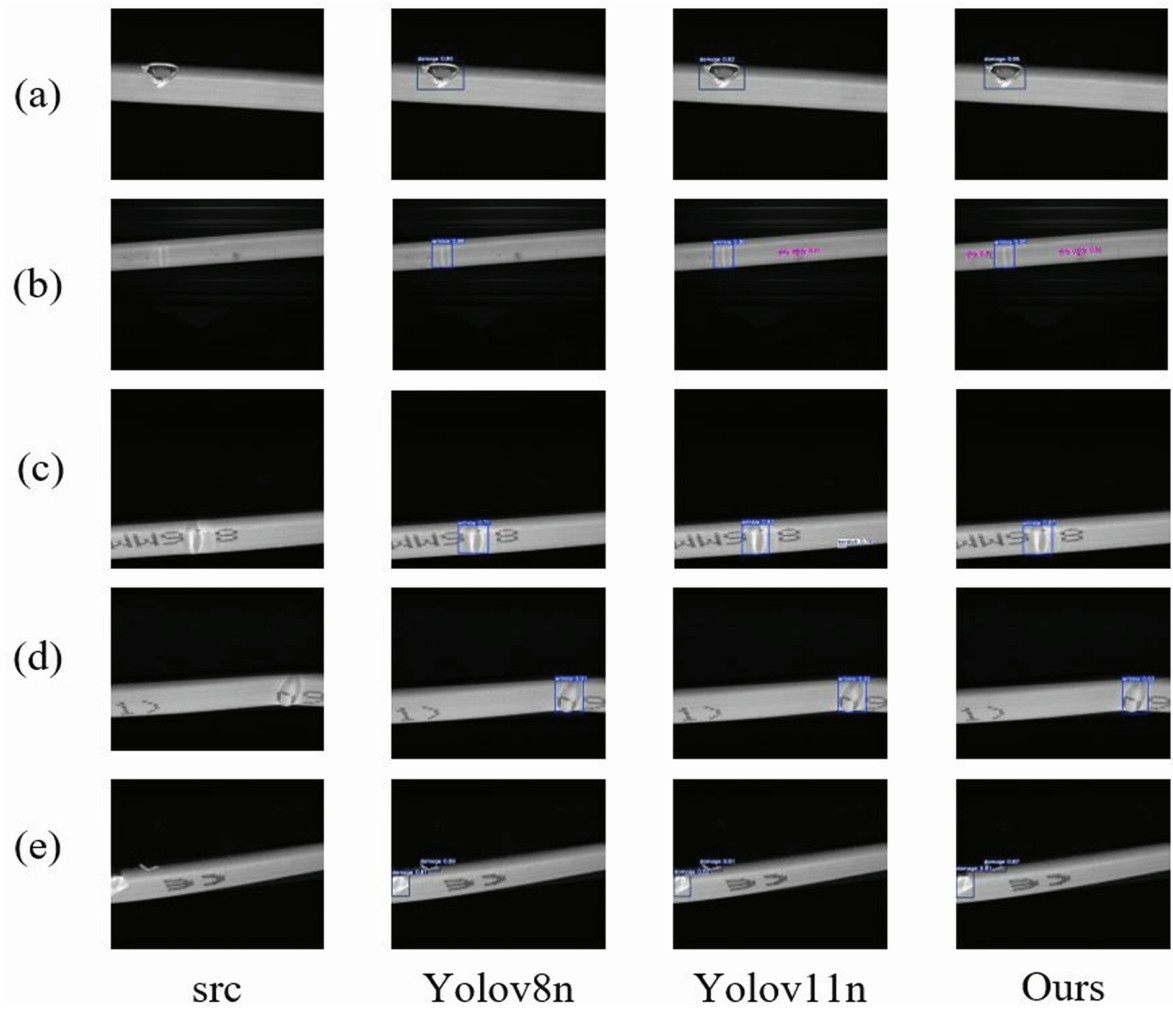

**Fig 7. Comparison of YOLOv8n, YOLOv11n and SAMF-YOLO.**

regular scenes with no significant occlusion or clutter. Both the baseline model YOLOv11s and SAMF-YOLO accurately detect the defective targets in these images. The following rows illustrate scenes with multiple defects. Although YOLOv11s occasionally misses some defect targets, its detection accuracy is relatively lower compared to SAMF-YOLO. In contrast, SAMF-YOLO demonstrates higher detection accuracy, successfully and precisely identifying all defect targets in the images. SAMF-YOLO excels at detecting and annotating small, subtle targets, highlighting its advantage in handling complex scenes. Overall, the results indicate that the improved SAMF-YOLO model shows higher detection accuracy and robustness in various complex scenarios, particularly excelling at detecting small and intricate defects, thus demonstrating outstanding performance in challenging environments.

Score-CAM [41] is a class activation mapping method based on feature map scoring, aimed at improving the transparency and interpretability of convolutional neural network models. Unlike traditional gradient-based methods, Score-CAM generates class activation maps by directly computing the activation strength of feature maps, without the need for gradient calculations or backpropagation information. This approach helps avoid issues like

gradient vanishing while providing more intuitive and precise visual explanations. Unlike previous class activation mapping methods [42,43], Score-CAM does not rely on gradients or weighting mechanisms; instead, it highlights key regions that influence decision-making by scoring the feature maps directly. This method not only enhances computational efficiency but also reduces noise interference, resulting in clearer and more understandable activation maps. Through this approach, Score-CAM effectively showcases regions in the input image closely related to the model's prediction, such as subtle defect areas.

Furthermore, Score-CAM exhibits high computational efficiency, as its simplified calculation process accelerates the generation of class activation maps. For real-time detection tasks, it demonstrates a significant advantage, making it particularly suitable for applications requiring high model interpretability and decision transparency. Due to these features, Score-CAM has become a powerful tool, enhancing the interpretability of the SAMF-YOLO model, aiding in better understanding its reasoning process, and providing strong support for the model's robustness and generalization capabilities.

Fig 8 shows the heatmap performance of the baseline model YOLOv11s and the improved SAMF-YOLO model in different scenarios. Each row represents a different scene: the first row shows a regular scene, the second row displays a scene with occlusion, and the third row shows a scene with small occluded targets. The left column displays the original image, the middle column shows the heatmap generated by YOLOv11s, and the right column shows the heatmap generated by SAMF-YOLO. In the regular scene, both YOLOv11s and SAMF-YOLO accurately identify the targets, but the SAMF-YOLO heatmap is more concentrated, reflecting greater attention to the target. For occluded scenes, the YOLOv11s heatmap shows some signal confusion and scattered focus, failing to effectively concentrate on the target. In contrast, SAMF-YOLO can accurately localize the target area, maintaining detection accuracy even in

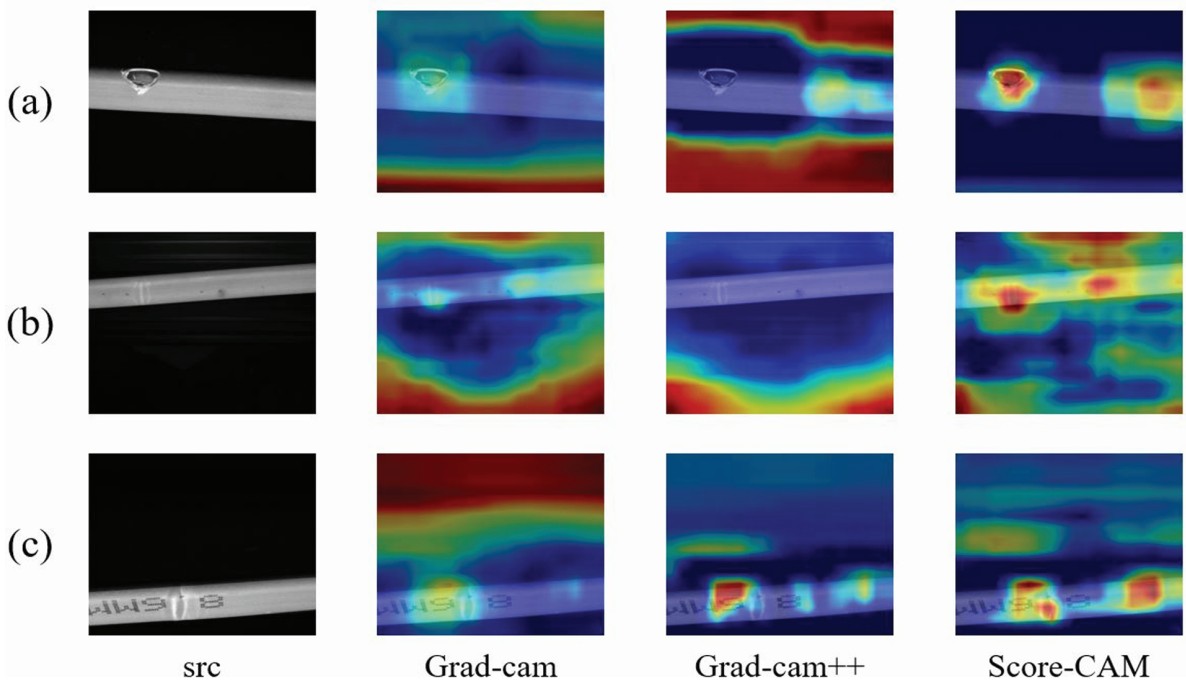

**Fig 8. Heatmap comparison.**

complex backgrounds. Finally, in the scene with small occluded targets, YOLOv11s misses the target that has fallen to the right, exhibiting weaker heatmap performance. In comparison, SAMF-YOLO maintains high brightness and focus, better emphasizing the target area in these challenging scenarios. Overall, SAMF-YOLO demonstrates significant advantages in handling subtle defects and small target detection, showcasing its stability and effectiveness across different levels of complexity.

## Conclusion

This study presents SAMF-YOLO, a lightweight yet accurate detector designed for industrial defect inspection. Compared with YOLOv11s, SAMF-YOLO achieves higher Precision (76.5%), Recall (74.8%), and mAP@0.5 (75.7%), while reducing GFLOPs by 41.8% and maintaining real-time speed at 77.9 FPS. These improvements stem from the effective integration of UniRepLKNet, C3K2-BFAM, FASFF-Head, and Focaler-IoU, each contributing to robust, efficient detection. Overall, SAMF-YOLO strikes a strong balance between performance and efficiency, making it well-suited for real-world industrial applications.

## Author contributions

**Conceptualization:** Jun Huang, Qun Yang.

**Data curation:** Qiang Zhu.

**Formal analysis:** Wanting Xu.

**Methodology:** Jun Huang.

**Project administration:** Jun Huang.

**Resources:** Jun Huang.

**Software:** Qun Yang.

**Supervision:** Shamsul Arrieya Ariffin.

**Validation:** Jun Huang.

**Writing – original draft:** Jun Huang.

**Writing – review & editing:** Jun Huang.

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
