## [Decision Letter · Decision Letter 0]

7 Apr 2025

PONE-D-25-10238SAMF-YOLO: A Self-Supervised, High-Precision Approach for Defect Detection in Complex Industrial EnvironmentsPLOS ONE

Dear Dr. Huang,

Thank you for submitting your manuscript to PLOS ONE. After careful consideration, we feel that it has merit but does not fully meet PLOS ONE’s publication criteria as it currently stands. Therefore, we invite you to submit a revised version of the manuscript that addresses the points raised during the review process.

We look forward to receiving your revised manuscript.

Kind regards,

Qian Zhang, Ph.D

Academic Editor

PLOS ONE

Journal Requirements:

Reviewers' comments:

Reviewer's Responses to Questions

**Comments to the Author**

1. Is the manuscript technically sound, and do the data support the conclusions?

Reviewer #1: Partly

Reviewer #2: Yes

2. Has the statistical analysis been performed appropriately and rigorously? 

Reviewer #1: Yes

Reviewer #2: Yes

3. Have the authors made all data underlying the findings in their manuscript fully available?

Reviewer #1: No

Reviewer #2: Yes

4. Is the manuscript presented in an intelligible fashion and written in standard English?

Reviewer #1: Yes

Reviewer #2: Yes

5. Review Comments to the Author

Reviewer #1: In the chapter "Experimental results and analyses Experimental environments and dataset", the author mentioned, "To validate the robustness and generalization capability of the SAMF-YOLO model, two distinct datasets were used in this study. The first dataset is a self-collected solar heating tape defect dataset, which includes 907 images for training and 227 images for validation. This dataset covers various defect types, including wrinkles, dents, scratches, punctures, damage, and dirt, making it suitable for multiple defect detection tasks. These defects provide a diverse set of scenarios for testing the model’s generalization performance in detecting anomalies across different objects and conditions, making it an ideal benchmark for evaluating anomaly detection systems." In this paragraph, I noticed that the author used two datasets. One is the self-collected dataset, while the other dataset was not introduced, and no link to the open-source dataset was provided. Also, neither the subsequent paragraphs nor the tables provided the performance of the model on the open-source dataset. Did the author combine the self-collected dataset with the open-source dataset and then re-divide the dataset?

The author should provide a detailed introduction to the self-collected dataset. More importantly, the author should introduce the open-source dataset used, cite the open-source dataset, or provide the link to the dataset. After conducting experiments on the self-collected dataset using the proposed method, the author should validate it on the open-source dataset, which would make the findings more convincing.

I think there are certain problems in the author's experimental section. The author should clearly introduce the datasets used, including the self-collected dataset and the open-source dataset, as well as the performance of the model on both the private dataset and the open-source dataset. It is recommended that the author make sufficient revisions to the experimental section.

Reviewer #2: Scientific Review Comments

1. Novelty and Contributions

- Strengths:

- The proposed SAMF-YOLO model integrates innovative components such as UniRepLknet, FASFF-Head, and BFAM, leveraging Star Operation, adaptive multi-scale fusion, and bi-temporal feature aggregation to significantly improve detection accuracy (6.38% increase in mAP@0.5) and computational efficiency (~75% reduction in parameters and GFLOPs) for small defect detection in complex industrial environments.

- The Focaler-IoU loss function addresses sample imbalance through linear interval mapping, demonstrating superior performance in bounding box regression compared to existing losses (Table 4).

- The incorporation of self-supervised contrastive learning reduces reliance on labeled data and enhances model robustness, aligning with practical industrial needs.

Concerns:

- Theoretical validation of Star Operation: The assumption that implicit feature space dimensions grow exponentially with layer depth (Equations 7–11) lacks empirical verification (e.g., correlation between feature dimensionality and performance).

- Incomplete self-supervised framework: The contrastive learning section lacks details on positive/negative pair generation strategies and loss optimization, hindering reproducibility.

2. Experimental Design and Results

-Strengths:

- Dataset diversity: Validation on a self-collected solar heating tape defect dataset (multiple defect types) and public datasets demonstrates generalization capability.

- Comprehensive evaluation: Ablation studies (Table 3) and comparisons with YOLO variants, RT-DETR (Table 5) are thorough and convincing.

- Visualization: Heatmaps (Fig. 8) and detection results (Fig. 7) effectively illustrate the model’s superiority in handling small targets and occluded scenes.

Concerns:

- Limited evaluation metrics: Only mAP@0.5 is reported; additional metrics (e.g., mAP@0.5:0.95, FPS) are needed for deployment feasibility analysis.

- Small-scale dataset: The self-built dataset (907 training images) may limit generalization. Validation on large-scale benchmarks (e.g., COCO) is recommended.

- Missing experimental details: Specifics on data augmentation (rotation angles, crop ratios) and hyperparameter tuning (learning rate, momentum) are unclear.

3. Methodological Rigor

- Strengths:

- Mathematical clarity: Equations for Star Operation (1–11) and Focaler-IoU (23–24) are logically derived with well-defined notation.

- Module design: BFAM’s multi-scale dilated convolutions and SimAM attention theoretically enhance small defect detection; FASFF-Head’s soft-weighted fusion (Equations 15–16) mitigates cross-scale conflicts.

- Concerns:

- Inconsistent notation: The implicit feature dimension derivation in Equation 4 (\(\frac{(d+2)(d+1)}{2}\)) conflicts with recursive formulas in Equations 7–11 (\(\left(\frac{d}{2}\right)^{2^l}\)). Mathematical consistency requires revision.

- BFAM complexity: Multi-branch operations (Equations 17–22) may increase computational load. Computational cost analysis for BFAM is missing.

4. Writing and Presentation

- Strengths:

- Structural coherence: Follows IMRAD format with clear logic. Figures (1–8) and tables (1–5) align with textual descriptions.

- Terminology standardization: Abbreviations (e.g., mAP@0.5, GFLOPs) are well-defined. References cover key works (2014–2024), ensuring timeliness.

- Improvements needed:

- Language optimization: Overuse of "while" in the abstract and inconsistent terminology (e.g., "BEAM" vs. "BFAM") should be corrected.

- Figure clarity: Fig. 1 (model architecture) lacks resolution; Fig. 6 is mislabeled (loss curves vs. BFAM details).

5. Ethics and Reproducibility

- Strengths:

- Data accessibility: Compliance with PLOS ONE policy via the statement: “All relevant data are within the manuscript and its Supporting Information files.”

- Ethical compliance: No human/animal experiments; conflicts of interest declared.

- Concerns:

- Code availability: Code repository links are absent, limiting reproducibility. GitHub access or supplementary code files are strongly recommended.

---

Recommendation

Decision: Conditional Acceptance (Major Revision)

Critical Revisions Required

1. Theoretical enhancements:

- Validate Star Operation’s implicit feature expansion via experiments (e.g., feature visualization or dimensionality-performance correlation).

- Elaborate on contrastive learning implementation (pair generation, loss hyperparameters).

2. Experimental expansion:

- Report mAP@0.5:0.95 and FPS for comprehensive evaluation.

- Test generalization on large-scale datasets (e.g., COCO).

3. Writing corrections:

- Resolve notation conflicts (Equations 4 vs. 7–11), terminology errors (BEAM→BFAM), and figure mislabeling.

- Optimize sentence structure and supplement experimental parameters.

4. Reproducibility:

- Release training code, pretrained models, and full data augmentation configurations.

If adequately addressed, this work holds significant potential to advance object detection in industrial applications.

6. PLOS authors have the option to publish the peer review history of their article (what does this mean?). If published, this will include your full peer review and any attached files.

Reviewer #1: No

Reviewer #2: No

---

## [Author Response · Author response to Decision Letter 1]

2 May 2025

Responses to Reviewers

Original Manuscript ID:PONE-D-25-10238

Original Article Title: “ A Self-Supervised, High-Precision Approach for Defect Detection in Complex Industrial Environments”

To: PLOS One Editor

Re: Response to reviewers

Dear Editor,

Thank you for allowing a resubmission of our manuscript, with an opportunity to address the reviewers’ comments.

We are uploading our point-by-point response to the comments (below) (response to reviewers), an updated manuscript with yellow highlighting indicating changes (as “Revised Manuscript with Track Changes”), and a clean updated manuscript without highlights (“Manuscript”).

Best regards,

JUN HUANG et al.

Reviewer #1:

Comment: In the chapter "Experimental results and analyses Experimental environments and dataset", the author mentioned, "To validate the robustness and generalization capability of the SAMF-YOLO model, two distinct datasets were used in this study. The first dataset is a self-collected solar heating tape defect dataset, which includes 907 images for training and 227 images for validation. This dataset covers various defect types, including wrinkles, dents, scratches, punctures, damage, and dirt, making it suitable for multiple defect detection tasks. These defects provide a diverse set of scenarios for testing the model's generalization performance in detecting anomalies across different objects and conditions, making it an ideal benchmark for evaluating anomaly detection systems." In this paragraph, I noticed that the author used two datasets. One is the self-collected dataset, while the other dataset was not introduced, and no link to the open-source dataset was provided. Also, neither the subsequent paragraphs nor the tables provided the performance of the model on the open-source dataset. Did the author combine the self-collected dataset with the open-source dataset and then re-divide the dataset?

The author should provide a detailed introduction to the self-collected dataset. More importantly, the author should introduce the open-source dataset used, cite the open-source dataset, or provide the link to the dataset. After conducting experiments on the self-collected dataset using the proposed method, the author should validate it on the open-source dataset, which would make the findings more convincing.

I think there are certain problems in the author's experimental section. The author should clearly introduce the datasets used, including the self-collected dataset and the open-source dataset, as well as the performance of the model on both the private dataset and the open-source dataset. It is recommended that the author make sufficient revisions to the experimental section.

Response:

Thank you very much for your valuable comments.

To address your concerns regarding the datasets and experimental design, we have made the following clarifications and revisions:

1. Detailed Introduction of Datasets: In the revised manuscript (page 10, Experimental results and analyses – Experimental environments and dataset), we have added a detailed description of the open-source dataset NEU-DET, including its categories, size, and relevance to defect detection tasks. The self-collected dataset is also described in more detail.

2. Dataset Citation and Accessibility: Although the NEU-DET dataset is publicly available, its download link is provided alongside the code repository link in the abstract(https://github.com/Missing24ff/SAMF-YOLO.git), ensuring accessibility for readers and reviewers.

3. Experimental Results on Both Datasets: We have conducted separate experiments on both datasets. The results of our method on NEU-DET are presented in Table 7 (page 14), where we compare SAMF-YOLO with other methods using multiple metrics, including Precision (%), mAP@0.5 (%), mAP@0.5:0.95 (%), GFLOPs, and FPS.

No Dataset Merging: We confirm that the self-collected dataset and the NEU-DET dataset are used independently for evaluation, and no merging or re-splitting was performed.

We hope these revisions sufficiently address your concerns, and we thank you again for your constructive feedback, which has helped us improve the completeness and clarity of our manuscript.

Reviewer #2:

Comment 1: Theoretical validation of Star Operation: The assumption that implicit feature space dimensions grow exponentially with layer depth (Equations 7-11) lacks empirical verification (e.g., correlation between feature dimensionality and performance).

Response:

Thank you for pointing out the lack of empirical validation regarding the theoretical assumption in the Star Operation.To address this issue, we have made the following revisions:

1. Theoretical Adjustment: We have revised Equation 4 and the original Equations 7–11 into a more compact recursive form (now Equation 7) to better reflect the theoretical formulation of the implicit feature space growth.

2.Empirical Validation: In the Ablation Experiments section (page 12), we have added new experiments to empirically evaluate the relationship between layer depth and the implicit feature space dimensionality in the Star Operation. These results are presented in Table 3, demonstrating how performance varies with the growth of the feature space.

We appreciate your insightful feedback, which has helped us strengthen the theoretical and experimental soundness of the proposed method.

Comment 2: Incomplete self-supervised framework: The contrastive learning section lacks details on positive/negative pair generation strategies and loss optimization, hindering reproducibility.

Response:

Thank you for your helpful comment regarding the contrastive learning framework.We have revised the manuscript (page 5) to include more details on this section. Specifically:

1.Positive and Negative Pair Strategy: We now clearly describe that positive pairs are generated by applying two different augmentations to the same image, while negative samples are formed by using other images in the same batch.

2.Loss Optimization: We clarify that a contrastive loss is used to optimize the similarity between positive pairs and dissimilarity with negative pairs, following common practices in self-supervised learning.

3.Training Settings: We also provide training details, including using the SGD optimizer with an initial learning rate of 0.01, momentum of 0.937, and weight decay of 5×10−4, along with a cosine annealing learning rate schedule.

These updates enhance the completeness and reproducibility of the self-supervised framework. Thank you again for your valuable feedback.

Comment 3: Limited evaluation metrics: Only mAP@0.5 is reported; additional metrics (e.g., mAP@0.5:0.95, FPS) are needed for deployment feasibility analysis.

Response:

Thank you for your valuable suggestion regarding the evaluation metrics.

In the revised manuscript, we have added additional evaluation metrics, including mAP@0.5:0.95 and FPS, to provide a more comprehensive assessment of the model's performance and its deployment feasibility. These results are now presented alongside mAP@0.5 in the updated experimental section.

We appreciate your feedback, which helped us improve the completeness and practical relevance of our evaluation.

Comment 4: Small-scale dataset: The self-built dataset (907 training images) may limit generalization. Validation on large-scale benchmarks (e.g., COCO) is recommended.

Response:

Thank you for your constructive comment regarding the dataset scale and generalization.

To enhance the robustness and generalization analysis of our method, we have added experiments on a publicly available industrial defect detection dataset, NEU-DET, in the Evaluation Metrics section. This complements the self-built dataset and provides a more reliable assessment of the model’s generalization capability on a larger-scale benchmark.

We appreciate your feedback, which helped us strengthen the experimental design of our work.

Comment 5: Missing experimental details: Specifics on data augmentation (rotation angles, crop ratios) and hyperparameter tuning (learning rate, momentum) are unclear.

Response:

Thank you for your helpful comment regarding the missing experimental details.

In the revised manuscript (page 5), we have added specific descriptions of the data augmentation parameters, including rotation angles, crop ratios, and probabilities for each transformation. Additionally, we have included the details of hyperparameter settings, such as the learning rate, momentum, weight decay, and the learning rate scheduling strategy.

These additions enhance the reproducibility and clarity of our experimental setup. We appreciate your feedback in helping us improve the completeness of the manuscript.

Comment 6: Inconsistent notation: The implicit feature dimension derivation in Equation 4 (\frac(d+2)(d+1))(2)V) conflicts with recursive formulas in Equations 7-11 (\frac(d)[2right)[2^|). Mathematical consistency requires revision.

Response:

Thank you for pointing out the inconsistency in the mathematical notation.

As mentioned earlier, we have revised Equation 4 and the original Equations 7–11, and unified them into a compact recursive formulation (now presented as Equation 7). This adjustment ensures consistency in the theoretical derivation of the implicit feature space dimensionality and aligns all related expressions under a coherent mathematical framework.

We appreciate your insightful feedback, which helped improve the clarity and rigor of the theoretical presentation.

Comment 7: BFAM complexity: Multi-branch operations (Equations 17-22) may increase computational load. Computational cost analysis for BFAM is missing.

Response:

Thank you for your insightful comment regarding the potential computational overhead introduced by the BFAM module.

We have conducted a detailed ablation study by integrating BFAM into the C3k2 layer. The results show that:

The parameter count increased by 0.6M.

The inference speed (FPS) decreased by approximately 1.8.

However, the detection accuracy improved significantly, with +2.16% mAP@0.5 and +1.3% mAP@0.5:0.95.

Given the performance gains, we believe the added computational cost introduced by BFAM is acceptable and justified in the context of industrial defect detection tasks.

We have included this analysis in the updated experimental results section to provide a more comprehensive understanding of BFAM's cost-effectiveness.

Comment 8: Language optimization: Overuse of "while" in the abstract and inconsistent terminology (e.g., "BEAM" vs. "BFAM") should be corrected.

Response:

Thank you for pointing out the language issues in the abstract and terminology usage.

We have carefully revised the abstract to reduce the overuse of "while" and improved the overall sentence structure for better readability. Additionally, we have corrected the inconsistent terminology and now consistently use "BFAM" throughout the manuscript.

We appreciate your feedback, which helped us enhance the clarity and consistency of the paper.

Comment 9: Figure clarity: Fig. 1 (model architecture) lacks resolution; Fig. 6 is mislabeled (loss curves vs. BFAM details).

Response:

Thank you for your helpful comment regarding the figure clarity and labeling.

We have updated Figure 1 with a higher-resolution version to improve its readability and visual clarity. Additionally, the mislabeling issue with Figure 6 has been corrected — it now accurately reflects the corresponding content (loss curves vs. BFAM details).

We appreciate your feedback, which helped us enhance the presentation quality of the manuscript.

Comment 10: Code availability: Code repository links are absent, limiting reproducibility. GitHub access or supplementary code files are strongly recommended.

Response:

Thank you for your valuable comment regarding code availability.

We have now included the open-source code repository link in the abstract, allowing readers to access the full implementation for reproducibility and further research. The code is hosted on GitHub and includes all necessary files and instructions for running the experiments.

We appreciate your feedback, which helped us improve the transparency and reproducibility of our work.

---

## [Decision Letter · Decision Letter 1]

9 Jun 2025

SAMF-YOLO: A Self-Supervised, High-Precision Approach for Defect Detection in Complex Industrial Environments

PONE-D-25-10238R1

Dear Dr. Huang,

We’re pleased to inform you that your manuscript has been judged scientifically suitable for publication and will be formally accepted for publication once it meets all outstanding technical requirements.

Kind regards,

Qian Zhang, Ph.D

Academic Editor

PLOS ONE

Additional Editor Comments (optional):

Reviewers' comments:

Reviewer's Responses to Questions

**Comments to the Author**

1. If the authors have adequately addressed your comments raised in a previous round of review and you feel that this manuscript is now acceptable for publication, you may indicate that here to bypass the “Comments to the Author” section, enter your conflict of interest statement in the “Confidential to Editor” section, and submit your "Accept" recommendation.

Reviewer #1: All comments have been addressed

Reviewer #2: All comments have been addressed

2. Is the manuscript technically sound, and do the data support the conclusions?

Reviewer #1: Yes

Reviewer #2: (No Response)

3. Has the statistical analysis been performed appropriately and rigorously? 

Reviewer #1: Yes

Reviewer #2: (No Response)

4. Have the authors made all data underlying the findings in their manuscript fully available?

Reviewer #1: Yes

Reviewer #2: No

5. Is the manuscript presented in an intelligible fashion and written in standard English?

Reviewer #1: Yes

Reviewer #2: No

6. Review Comments to the Author

Reviewer #1: During the re-review process, I carefully examined all aspects of the paper, including the methodology, experiments, and results. The authors have adequately addressed the issues I raised previously and provided satisfactory answers. However, there are still some minor issues. The review of computer vision in the Introduction section of the paper is insufficient. It is suggested to supplement with the latest references:

(a) An Enhanced Deep Learning Model for Effective Crop Pest and Disease Detection

(b) PSSCL: A progressive sample selection framework with contrastive loss designed for noisy labels

(c) BPT-PLR: A Balanced Partitioning and Training Framework with Pseudo-Label Relaxed Contrastive Loss for Noisy Label Learning

(d) WSS-YOLO: An improved industrial defect detection network for steel surface defects

Reviewer #2: (No Response)

7. PLOS authors have the option to publish the peer review history of their article (what does this mean?). If published, this will include your full peer review and any attached files.

Reviewer #1: No

Reviewer #2: No

---

## [Editor Report · Acceptance letter]

PONE-D-25-10238R1

PLOS ONE

Dear Dr. Huang,

I'm pleased to inform you that your manuscript has been deemed suitable for publication in PLOS ONE. Congratulations! Your manuscript is now being handed over to our production team.

Kind regards,

on behalf of

Dr. Qian Zhang

Academic Editor

PLOS ONE